# Genetically Modified Circulating Levels of Advanced Glycation End-Products and Their Soluble Receptor (AGEs-RAGE Axis) with Risk and Mortality of Breast Cancer

**DOI:** 10.3390/cancers14246124

**Published:** 2022-12-12

**Authors:** Yu Peng, Fubin Liu, Yating Qiao, Peng Wang, Han Du, Changyu Si, Xixuan Wang, Kexin Chen, Fangfang Song

**Affiliations:** Department of Epidemiology and Biostatistics, Key Laboratory of Molecular Cancer Epidemiology, Key Laboratory of Breast Cancer Prevention and Therapy, Tianjin Medical University, Ministry of Education, National Clinical Research Center for Cancer, Tianjin’s Clinical Research Center for Cancer, Tianjin Medical University Cancer Institute and Hospital, Tianjin 300060, China

**Keywords:** advanced glycation end-products, soluble receptor for advanced glycation end-products, breast cancer, genetic polymorphism

## Abstract

**Simple Summary:**

The large-scale population-based evidence exploring genetically modified circulating levels of advanced glycation end-products (AGEs) and their receptor (RAGE) with risk and mortality of breast cancer is scarce. We aimed to evaluate the role of plasma AGEs, soluble RAGE (sRAGE), and AGEs/sRAGE-ratio level, as well as their interactions with a genetic predisposition in their metabolism-related genes on breast cancer. Higher levels of AGEs and AGEs/sRAGE-ratio were associated with an increased risk of breast cancer, but sRAGE levels were negatively associated with breast cancer risk. We also observed a positive association between AGEs and the bad prognosis of breast cancer. Although we did not observe a significant contribution of genetic variants to breast cancer risk, rs2070600 and rs1800624 in the *AGER* gene were dose-dependently correlated with sRAGE levels. Further, compared to the haplotype CT at the lowest quartile of AGEs, haplotypes TT and TA were prominently associated with breast cancer risk in the highest quartile of AGEs. This study emphasized the potential of the AGE-RAGE axis as a new biomarker of breast cancer in the future and aided in screening a high-risk population with breast cancer.

**Abstract:**

The interaction of advanced glycation end-products (AGEs) with their receptor (RAGE) elicits oxidative stress and inflammation, which is involved in the development of breast cancer. However, large-scale population-based evidence exploring genetically modified circulating levels of AGEs-RAGE axis with risk and mortality of breast cancer is scarce. We recruited 1051 pairs of age-matched breast cancers and controls and measured plasma AGEs and sRAGE concentrations by enzyme-linked immunosorbent assay (ELISA). Multivariate logistic regression and Cox proportional hazard model were used to calculate the effects of plasma levels and genetic variants of the AGEs-RAGE axis and their combined effects on breast cancer risk and prognosis, respectively. Furthermore, linear regression was performed to assess the modifications in plasma AGEs/sRAGE levels by genetic predisposition. Higher levels of AGEs and AGEs/sRAGE-ratio were associated with an increased risk of breast cancer, but sRAGE levels were negatively associated with breast cancer risk, especially in women <60 years. We also observed a positive association between AGEs and the bad prognosis of breast cancer. Although we did not observe a significant contribution of genetic variants to breast cancer risk, rs2070600 and rs1800624 in the *AGER* gene were dose-dependently correlated with sRAGE levels. Further, compared to the haplotype CT at the lowest quartile of AGEs, haplotypes TT and TA were prominently associated with breast cancer risk in the highest quartile of AGEs. This study depicted a significant association between circulating levels of AGEs-RAGE axis and breast cancer risk and mortality and revealed the potential of plasma AGEs, especially coupled with *AGER* polymorphism as biomarkers of breast cancer.

## 1. Background

Breast cancer is a serious threat to global women’s health. In 2020, there were around 2.26 million new cases and 680 thousand new deaths of breast cancer worldwide [1]. Many studies suggest that genetic factors and modifiable dietary and lifestyle risk factors contribute largely to the occurrence of breast cancer [2,3]. Advanced glycation end-products (AGEs), a heterogeneous group of molecules formed by non-enzymatic reactions between reducing sugars and proteins, lipids, or nucleic acids, are endogenous by-products of normal metabolism [4]. However, it is worth noting that some breast cancer-related risk factors, such as aging, exogenous unhealthy diet, smoking, and a sedentary lifestyle, will also lead to the irreversible accumulation of AGEs in tissues [4,5,6,7]. This will possess pathogenic effects on organ homeostasis, genetic integrity, and cellular signaling cascades [5], which may lead to the occurrence of breast cancer. At present, more than 20 different AGEs have been found in human blood, tissues, and food, such as N (6)-carboxymethyl-lysine (CML) and carboxyethyl lysine (CEL) [8]. Due to the diversity of AGEs, the database of dietary AGEs cannot accurately estimate the total exposure level of AGEs and reflect the comprehensive exposure of exogenous AGEs as well as the overall burden of the body [9,10,11]. Alternatively, blood AGEs levels may represent a sensitive marker for assessing endogenous and exogenous exposure levels of AGEs in the organism [8,12]. However, population-based epidemiological study on the relationship between blood levels of AGEs and breast cancer was scarce and limited by a small sample size [5].

The metabolism and activation of AGEs in vivo are mediated by many cell surface receptors, such as AGEs receptor 1/2/3 (AGE-R1/2/3) [7,12] and receptor for AGEs (RAGE) [6]. The pathogenic effect of AGEs is mainly through the activation of RAGE [6,8,13], and overexpression of RAGE has been confirmed in breast cancer and many other cancer tissues [11]. In addition to the full-length RAGE, a soluble receptor for advanced glycation end-products (sRAGE) [14] can exert a protective effect on the organism by preventing the interaction of AGEs and RAGE [4,14,15]. Therefore, a comprehensive evaluation of AGEs and sRAGE levels (such as AGEs/sRAGE-ratio) is more valuable for exploring the potential role of the AGEs-RAGE axis in diseases [16]. Only a few small-sample studies suggested sRAGE might be a new potential biomarker for predicting breast, lung, and colorectal cancers [17], let alone the large-scale population evidence on AGEs-RAGE axis levels with breast cancer risk. In addition, single-nucleotide polymorphisms (SNPs) of these receptor genes, such as the *AGER* gene encoding RAGE, have been found to be associated with multiple chronic diseases [18,19]. However, the role of individual genetic variations involved in the AGEs-RAGE axis and how they interact with AGEs/sRAGE levels in the development of breast cancer remains unclear.

The aim of the present study was to evaluate the role of plasma AGEs, sRAGE, and AGEs/sRAGE-ratio level, as well as their interactions with a genetic predisposition in their metabolism-related genes on breast cancer. The results will have great significance for providing a theoretical basis to explore new breast cancer markers for early diagnosis and personalized prevention of breast cancer.

## 2. Methods

### 2.1. Study Population and Data Collection

We consecutively recruited 1051 patients from the Tianjin Medical University Cancer Institute and Hospital, China. The inclusion criteria of cases were: (1) the residents in Tianjin aged 18~65, newly diagnosed and histologically confirmed breast cancer from 1 January 2003; (2) without treatment before admission for first-diagnosed cancer; (3) no history of other malignant tumors in the past; and (4) no history of blood transfusion 6 months before blood sampling. Concomitantly, we recruited 1051 healthy controls who were frequency-matched to patients by age from a population undergoing health check-ups in Tianjin Community. Finally, 1051 case-control pairs were recruited; after excluding 5 subjects whose plasma AGEs or sRAGE levels were not measured, 2039 participants without extreme values (beyond the range of mean ± 3 standard deviations) of plasma AGEs or sRAGE levels remained ultimately in the analysis (Appendix A). The Ethics Committee of Tianjin Medical University Cancer Institute and Hospital approved the study protocol, and written informed consent was obtained from all patients and controls to participate in this study.

The demographic and epidemiological data of participants were collected by a structured questionnaire. Medical records were retrieved to obtain detailed clinical and pathological data, as well as follow-up information (up to 31 August 2021). Peripheral blood samples (5~10 mL whole blood) were collected, and plasma and leukocytes were isolated respectively, then deoxyribonucleic acid (DNA) was extracted from leukocytes by the QIAquick polymerase chain reaction (PCR) Purification Kit (QIAGEN). The plasma and genomic DNA stock solution were stored in a refrigerator at −80°C for standby.

### 2.2. Measurement of Plasma AGEs and sRAGE Levels

Plasma AGEs and sRAGE concentrations were measured by enzyme-linked immunosorbent assay (ELISA) kit (Cell Biolabs, Inc., San Diego, CA, USA and Quantikine, R&D Systems, Inc., Minneapolis, MN, USA), following the manufacturer’s instructions and adopting blind test to avoid bias (the sample information of case-control is not disclosed to the tester). Intra- and inter-batch coefficients of variation (CV) were assessed by measuring 3 different samples used as quality controls in duplicate in each. Mean intra- and inter-batch CVs were 5.7% and 7.7%, respectively.

### 2.3. Selection of SNPs and DNA Genotyping

Referring to the public database and published literature, we selected 14 potential functional SNPs located in genes encoding receptors of AGEs (*RAGE, AGE-R1, AGE-R2,* and *AGE-R3*) with minimum allele frequency (MAF) ≥0.05 in Chinese for genotyping (Appendix A). SNPs were genotyped by Illumina GSA chip and TaqMan SNP typing technology (ABI’s fluorescence quantitative PCR system).

### 2.4. Statistical Analyses

Continuous variables were expressed as mean ± standard deviation (SD) or median [interquartile range (IQR)], and categorical variables were expressed as quantity and percentage. Quantitative variables were compared by Student’s *t*-test, Wilcoxon rank sum test, and Kruskal--Wallis test. The comparison of categorical variables was performed by Pearson’s chi-square test or Fisher’s exact test. We observed the distribution of AGEs and sRAGE by different clinical information and baseline characteristics. The relationship between AGEs and sRAGE as continuous variables and hormones was explored by Spearman’s rank correlation analysis.

Multivariate logistic regression analysis was applied to examine odds ratios (ORs) and 95% confidence intervals (CIs) to assess the association of AGEs, sRAGE, AGEs/sRAGE-ratio with risks of breast cancer, and different molecular subtypes of breast cancer. The AGEs, sRAGE, and AGEs/sRAGE-ratio were analyzed both as a categorical variable in quartiles (Q1-Q4, with the first quartile, Q1, as the reference category) and as a continuous variable, divided by its SD. The receiver operating curve (ROC), area under the curve (AUC), net reclassification improvements (NRI), and integrated discrimination improvement (IDI) were used to assess the ability of AGEs-RAGE axis and traditional risk factors to distinguish subjects with or without breast cancer. The nonlinear associations of the AGEs-RAGE axis with the risk of breast cancer were presented by using the restricted cubic spline (RCS) model. Subgroup analyses were performed according to body mass index (BMI), menopausal status, and age group, and the *P* value of interaction in subgroup analysis was estimated by a multivariable logistic regression model. Based on Schoenfeld residual analysis, the proportional risk assumption was satisfied. The Kaplan--Meier method and multivariate Cox proportional hazards model were used to explore the relationship between the AGEs-RAGE axis and breast cancer prognosis, and results showed hazard ratios (HRs) and 95% CIs.

Finally, the distribution of observed and anticipated genotype frequencies was compared using a Chi-squared test to investigate whether the genotype was in Hardy-Weinberg equilibrium. The distribution of AGEs and sRAGE by SNPs genotypes was observed in case-control, then we performed linear regression to explore the influence of SNPs on AGEs or sRAGE levels. To evaluate the joint effects of the SNPs associated with AGEs or sRAGE levels, haplotypes were constructed using Shapeit2. We used multivariable logistic regression and Cox proportional hazard model to assess the impact of the genotypes and interaction of haplotypes with AGEs on breast cancer risk and prognosis.

All analyses were performed using SAS version 9.4 (SAS Institute, USA) and R software (The R Foundation, http://www.r-project.org, accessed on 1 June 2022, version 4.0.2). A level of <0.05 for two-sided *p* values was considered statistically significant.

## 3. Results

### 3.1. Baseline Characteristics

The total study population consisted of 2039 subjects, including 1018 breast cancer patients and 1021 healthy controls (Table 1). Significantly larger proportions of patients than the controls possessed the well-known traditional risk factors of breast cancer, such as obesity, premenopausal, younger at menarche, history of benign breast disease, family history of cancer, etc. The concentrations of AGEs and AGEs/sRAGE-ratio in breast cancer patients with different molecular subtypes (luminal A, luminal B, HER2+, and basal-like) were higher than that in the control group, whereas breast cancer patients had lower sRAGE levels than controls, although we did not observe differences among these molecular subtypes (Table 1, Appendix A).

For baseline data, higher quartiles of AGEs levels were more frequently observed in people with obesity, lower education, smoking, premenopausal state, and negative events, whereas the higher quartiles of sRAGE levels were less frequently in obese, low educated, and premenopausal people (Table 2).

### 3.2. Association of Plasma Levels of AGEs-RAGE Axis with Risks of Breast Cancer

As shown in Figure 1A, after adjustment for confounding factors, the Q2, the Q3, and the Q4 of AGEs levels were associated with a significant increase in breast cancer risk compared with the Q1, with the highest risk of breast cancer in the Q4 of AGEs [OR (95% CI) = 5.421 (3.896–7.543)]. Concomitantly, the risk of breast cancer was prominently increased by 63.50% per SD increase in the AGEs. In contrast, patients with a higher sRAGE level had a lower risk of breast cancer [the highest vs. the lowest sRAGE quartile of OR (95% CI): 0.421(0.307–0.815)], and the risk was reduced by 27% per SD sRAGE increment. Moreover, with the increase of AGEs/sRAGE-ratio, the risk of breast cancer increased, similar to the results of AGEs. Considering different molecular subtypes (Appendix A), when compared to Q1 of AGEs, the risk of luminal A, luminal B, and HER2+ subtypes in the Q2, Q3 and Q4 gradually increased, and the risks increased by 68.70%, 67.40%, and 55.00% respectively for per SD increment by AGEs. With respect to the Basal-like subtype, the augmented risk of breast cancer was reported in the Q3 and Q4 of AGEs, then for per SD increment in AGEs, the risk increased by 42.60%. Conversely, the risks of luminal A and HER2+ in Q3 and Q4, as well as luminal B in Q4, were reduced, taking the first quartile of sRAGE as the reference. The risk of luminal A, luminal B, HER2+, and Basal-like decreased by 24.90%, 27.50%, 32.70%, and 27.60%, respectively, per SD increment in sRAGE levels. For the AGEs/sRGAE-ratio, the results were similar to AGEs for both categorical and continuous variables.

We used RCS to flexibly model and visualize the relation of AGEs and sRAGE with breast cancer (Figure 1B–D) and observed that AGEs and AGEs/sRAGE-ratio had a non-linear positive correlation with the risk of breast cancer (*P*_nonlinear_ < 0.001), which increased with higher concentrations of AGEs and AGEs/RAGE-ratios, with the risk reaching plateaus at 7.1 ng/mL of AGEs. On the contrary, higher sRAGE was linearly associated with a substantially decreased risk of breast cancer (*P*_nonlinear_ = 0.114).

We further compared the ability of the AGEs-RAGE axis to distinguish patients and controls. AGEs, sRAGE, and AGEs/sRAGE-ratio were used as predictors with an AUC of 0.672, 0.607, and 0.697, respectively (Appendix A). In addition, the AUCs obtained by using traditional risk factors alone, the combination of traditional risk factors and AGEs, AGEs/sRAGE-ratio were 0.723, 0.754, and 0.770, respectively (Appendix A). The NRI and IDI values further indicated that AGEs or AGEs/sRAGE-ratio had higher predictive power for the risk of breast cancer than the risk factors (Appendix A).

### 3.3. Subgroup Analysis

Subgroup analyses were conducted according to BMI, menopausal status, and age group (Table 3). The effects of the AGEs-RAGE axis on the risk of breast cancer were not prominently different among three groups with BMI < 23.9, overweight and obese, and there was no interaction between AGEs, sRAGE, AGEs/sRAGE-ratio, and BMI. It seemed that AGEs and AGEs/sRAGE-ratio had a higher risk of breast cancer in premenopausal women than those postmenopausal women, and the interactions between AGEs and menopausal status were marginally significant (*p* = 0.070). After stratifying by age, we could observe that the effects of AGEs, sRAGE, and AGEs/sRAGE-ratio in women younger than 60 were significantly greater than those older than 60, and there was a significant interaction between sRAGE and age group (*p* = 0.019).

### 3.4. Association of AGEs and sRAGE Levels with Clinicopathological Features and Prognosis of Breast Cancer Patients

The distributions of AGEs and sRAGE quartiles in tumor node metastasis (TNM) stage, subtypes of breast cancer, and breast densities were not significantly diverse (Appendix A). However, the highest quartile of AGEs had a higher proportion of deaths than the lower quartiles (*p* = 0.028). Compared with the first quartile, estradiol (E2) and luteinizing hormone (LH) concentrations increased in all the higher quartiles of AGEs (*p* = 0.025) and sRAGE (*p* = 0.001), respectively. The subsequent correlation analysis demonstrated that follicle-stimulating hormone (FSH) was negatively correlated with AGEs, while E2 and LH were positively correlated with AGEs and sRAGE, respectively (Appendix A).

Compared with the Q1 of AGEs, the Q3 and Q4 increased all-cause mortality of breast cancer (log-rank *p* = 0.027; Figure 2A). The survival curves among different sRAGE and AGEs/sRAGE-ratio quartiles were not statistically significant (both log-rank *p* values > 0.05, Figure 2B,C). In the multivariate Cox proportional hazards regression model (Figure 2D), the risk of all-cause mortality in breast cancer patients increased at Q3 and Q4 of AGEs, referring to the Q1 of AGEs, and the risk increased by 22.10% for 1-SD AGEs increment. Similar results were also observed for the Q3 vs. Q1 HR in sRAGE and AGEs/sRAGE-ratio.

### 3.5. Analyses of Genetic Variants

The minimum allele frequency of all SNPs involved in this study was comparable to the public database (Appendix A). The chi-squared test revealed that all SNPs did not deviate from Hardy–Weinberg equilibrium, except rs184003 and rs10916846 (Appendix A). There was no statistically significant difference between the case and control in the distribution of different SNPs genotypes, and no SNPs were found to be associated with breast cancer risk and prognosis after the adjustment for covariates (Appendix A). Further linear regression models assessing the relationships of the independent genetic variants with AGEs (Appendix A) and sRAGE levels (Appendix A and Figure 3A) demonstrated the concentration of sRAGE was significantly negatively associated with the minor allele (T) of SNP rs2070600 and positively associated with the minor allele (T) of SNP rs1800624 located in *AGER* gene in a dose-dependent manner both for the case (*p* < 0.001, *p* = 0.004, respectively) and control group (*p* < 0.001, *p* < 0.001, respectively). Weak linkage disequilibrium between these two *AGER* SNPs was observed (*D’* = 0.98; *r^2^* = 0.05), and we subsequently conducted haplotype analysis to better understand their roles. All the *AGER* haplotypes were not associated with breast cancer risk, except that haplotype CA was associated with a reduced risk of all-cause mortality when haplotype CT was used as the reference (Appendix A). Nevertheless, compared to the haplotype CT, there was a negative correlation between haplotypes CA, TA, and TT (alleles in order of rs2070600, rs1800624) and levels of sRAGE (Appendix A). In a joint analysis to investigate the combined association of *AGER* haplotypes and AGEs with the risk and prognosis of breast cancer, compared to haplotype CT (supposed to be the highest quartile of sRAGE) at Q1 of AGEs, haplotypes CT, CA, TT, and TA at Q3 and Q4 of AGEs were significantly associated with increased breast cancer risk gradually, particularly haplotype TA and TT at Q4 of AGEs (Figure 3B), but not with breast cancer prognosis (Appendix A).

## 4. Discussion

This large case-control study suggested that higher plasma levels of AGEs and AGEs/sRAGE-ratio were associated with increased risk of breast cancer, but sRAGE levels were inversely associated with breast cancer risk, especially for individuals aged <60 years, which may provide new potential biomarkers for breast cancer diagnosis. AGEs levels can also predict the prognosis of breast cancer. Moreover, *AGER* gene SNPs rs2070600 and rs1800624 were correlated with plasma sRAGE level, and the estimated haplotypes TT and TA were remarkably associated with breast cancer risk at the higher quartile of AGEs.

The endogenous AGEs generation in the process of glucose metabolism and the accumulation of exogenous AGEs from unhealthy diets and lifestyles are not only risk factors for breast cancer but also the largest contributors to the overall AGEs pool in the human body. This may lead to earlier aging, earlier disease onset, and deterioration [20]. In this study, the levels of AGEs were higher in smokers, obesity, low education, premenopausal state, and people who experienced negative events, as were traditional risk factors for breast cancer, suggesting AGEs may account for a biological link between cancer, environmental, and socioeconomic factors involved in cancer promotion.

At present, only three prospective cohort studies demonstrated an increased incidence or mortality risk of breast cancer ascribed to higher dietary AGEs intake based on the food frequency questionnaire (FFQ) [9,10,11]. These studies are all based on the dietary AGEs database currently available mainly for dietary CML, and thus are difficult to accurately assess total AGEs exposure level [21,22]. Alternately, detecting the levels of AGEs in plasma can more accurately evaluate the endogenous and exogenous load of AGEs. Only two small-scale clinical cross-sectional studies have shown that breast cancer patients had a high accumulation of AGEs or CML in cancer tissue [5,23] and elevated serum AGEs level [5], consistent with our findings. Although the combination of AGEs with risk factors for breast cancer did not significantly elevate AUC in the present study, our research from NRI and IDI may have provided a new angle at this study by demonstrating that AGEs could correctly reclassify some participants. Tumor cells are characterized by metabolism reprogramming to meet their high metabolic demand, and the well-known Warburg process of aerobic glycolysis allows cancer cells to adapt to higher glucose uptake and thus leads to the synthesis of AGEs and the subsequent cancer progression [17,24]. Unsurprisingly, in this study, plasma AGEs were positively correlated with a bad prognosis of breast cancer. Moreover, we found that AGEs were negatively and positively associated with FSH and E2, respectively, similar to previous studies, which reported high AGEs suppressed FSH and LH [25,26] Additionally, a positive association between sRAGE and LH was observed. However, our hormone-related research was conducted only in breast cancer patients, and thus the effect of AGEs and sRAGE on hormone levels was necessitated to be elucidated by further studies.

RAGE, a multi-ligand cell surface protein receptor belonging to the immunoglobulin superfamily, can recognize and bind to AGEs, contributing to the activation of key signaling pathways [4,12,27]. This can cause the high expression of cytokines, growth factors, and adhesion molecules, which will lead to the recruitment of immune cells and the induction of an inflammatory response to initiate the occurrence of cancer [4,8,12]. RAGE has different subtypes, among which sRAGE can bind to AGEs, but fails to stimulate intracellular signal transduction, thus preventing inflammation and oxidative stress [13,15]. Many studies have shown that plasma levels of sRAGE are associated with a modest reduction in risk of multiple cancers, e.g., digestive system cancers and breast cancer [28,29,30,31] confirmed by the inverse association between plasma sRAGE and risk of breast cancer observed in our large sample study. Currently, the association between sRAGE and mortality of breast cancer has not been studied. In our study, only the third quartile of sRAGE was positively associated with the risk of all-cause mortality in breast cancer. Thus, the relationship of sRAGE with breast cancer mortality appeared to be complex and remained to be elucidated [32]. In addition, it has been shown that the AGEs/sRAGE-ratio serves as a better biomarker for age-related diseases (diabetes and cardiovascular disease, etc.) than AGEs or sRAGE alone [16,33,34]. Consistently, the AGEs/sRAGE-ratio had a stronger effect on the risk of breast cancer compared with AGEs and sRAGE in this study.

We did not observe a connection between AGEs-RAGE axis-related SNPs and breast cancer risk, consistent with a previous genotyping study of RAGE gene polymorphisms in the Han Chinese population in Northeast China [35], except the rs1800624 polymorphism related to breast cancer risk in two Chinese studies with relatively small sample sizes (1042 and 398 subjects, respectively) [36,37]. Consistently, the study herein also showed that the major allele (C) of rs2070600 was closely associated with higher levels of sRAGE in multiple disease [38,39], while the literature on the relationship between the genotype of rs1800624 and sRAGE was inconsistent [37,40,41]. These need to be explored by more research. The haplotypes CA, TT, and TA constructed by these two functional SNPs (rs2070600 and rs1800624) were negatively associated with sRAGE, but it is not with breast cancer risk in our results. However, a combined analysis of haplotypes and AGEs revealed that, compared to haplotype CT at the lowest quartile of AGEs, haplotype TT and TA at the highest of AGEs were significantly linked to increased breast cancer risk. Thus, haplotypes TT and TA can be combined with AGEs levels to identify individuals at high risk of breast cancer.

There are several strengths in the present study. Firstly, this was the first and largest study to comprehensively evaluate the potential role of the AGEs-RAGE axis in breast cancer. Secondly, we investigated the interaction between AGEs-RAGE axis receptor-related SNPs and AGEs levels in a Chinese population in the development of breast cancer. However, several limitations of this study merited special consideration. Our study had the inherent limitations of a case-control study, such as recall bias and selection bias. Second, the primary population studied in this study was citizens of Tianjin (a Northern City in China), which may not be representative of the entire Chinese population, and a validation cohort was lacking. Finally, the study may be underpowered to demonstrate the effect of genetic variation in AGEs-RAGE axis receptor-related genes in the development of breast cancer due to the relatively smaller sample size for genetic analysis.

## 5. Conclusions

In conclusion, our findings provided the first comprehensive depiction of a significant association between plasma levels of the AGEs-RAGE axis and breast cancer risk in a Chinese population, especially in women aged <60 years which requires special attention. Higher levels of AGEs had a significantly positive effect on breast cancer risk in the population carrying haplotypes TA and TT (SNPs rs2070600 and rs1800624). Therefore, this study emphasized the potential of the AGE-RAGE axis as a new biomarker of breast cancer in the future and aided in screening a high-risk population with breast cancer.

## Figures and Tables

**Figure 1 cancers-14-06124-f001:**
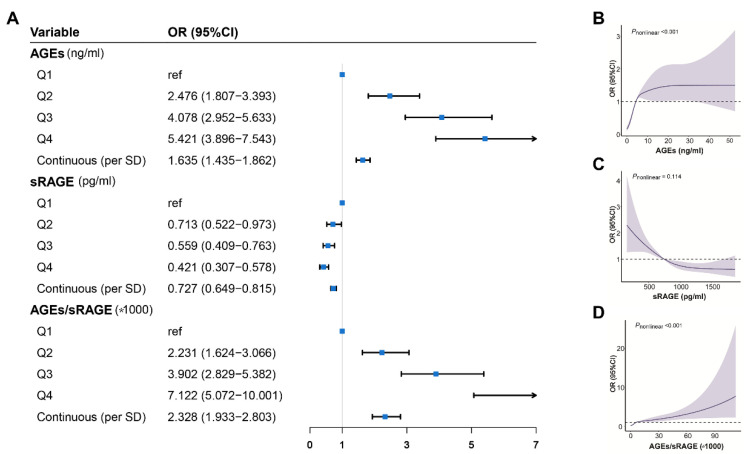
Associations and restricted cubic splines of plasma levels of AGEs-RAGE axis with risk of breast cancer. (**A**) Associations of plasma levels of AGEs-RAGE axis with risk of breast cancer. Restricted cubic splines of plasma levels of AGEs (**B**), RAGE (**C**), and AGEs/sRAGE-ratio (**D**) with risk of breast cancer. Covariates: body mass index, education, menarche age, menopause, estrogen replacement therapy, smoking, drinking, negative events, history of benign breast disease, breast cancer history of first-degree relatives. Abbreviations: AGEs, advanced glycation end-products; CI, confidence interval; OR, odds ratio; SD, standard deviation; sRAGE, soluble receptor for advanced glycation end-products.

**Figure 2 cancers-14-06124-f002:**
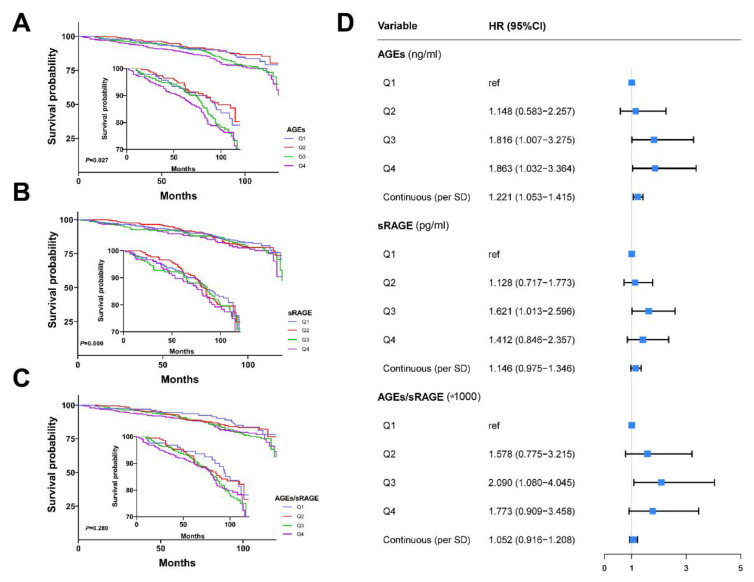
Associations between plasma levels of AGEs-RAGE axis and risk of all-cause mortality in breast cancer. Survival curves of AGEs (**A**), sRAGE (**B**), and AGEs/sRAGE-ratio (**C**) for breast cancer prognosis. (**D**) Cox proportional hazards models for the association between AGEs-RAGE axis and mortality of breast cancer. Covariates: body mass index, education, income, menopause, smoking, drinking, negative events, TNM stage, molecular subtype, cardiovascular disease, diabetes. Abbreviations: AGEs, advanced glycation end-products; CI, confidence interval; HR, hazard ratio; sRAGE, soluble receptor for advanced glycation end-products; TNM, tumor node metastasis.

**Figure 3 cancers-14-06124-f003:**
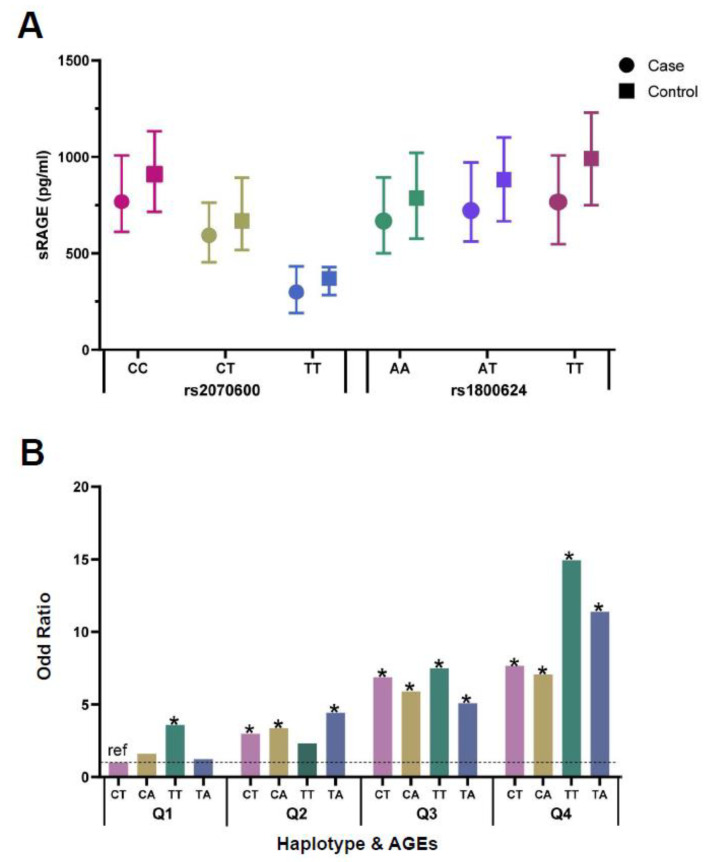
Associations of (**A**) SNPs rs2070600 and rs1800624 with sRAGE levels and (**B**) haplotype-AGEs interactions on breast cancer. Covariates for (**A**): body mass index, education, smoking, drinking, negative events. Covariates for (**B**): body mass index, education, menarche age, menopause, estrogen replacement therapy, smoking, drinking, negative events, history of benign breast disease, breast cancer history of first-degree relatives. Abbreviations: AGEs, advanced glycation end-products; ROC, receiver operating curve; sRAGE, soluble receptor for advanced glycation end-products. *: Compared with CT genotypes in the first quartile of AGEs, *p* < 0.05.

**Table 1 cancers-14-06124-t001:** Baseline characteristics of the case-control study.

Characteristics	Control (*N* = 1021)	Case (*N* = 1018)	*p*
Age (Mean ± SD)	53.6 ± 5.6	53.6 ± 6.1	0.925
BMI			<0.0001
≤23.9	494 (49.1)	397 (39.2)	
24–27.9	393 (39.1)	420 (41.5)	
≥28	119 (11.8)	195 (19.3)	
Education			<0.0001
Under primary school	40 (4.0)	185 (19.0)	
Junior/senior high school	734 (72.9)	614 (63.0)	
Junior college or above	233 (23.1)	176 (18.0)	
Smoke			0.044
No	873 (91.7)	891 (89.0)	
Yes	79 (8.3)	110 (11.0)	
Drinking			0.013
No	915 (95.81)	975 (97.79)	
Yes	40 (4.19)	22 (2.21)	
Negative events			<0.0001
No	869 (89.6)	656 (68.2)	
Yes	101 (10.4)	306 (31.8)	
Menarche			
≤13	170 (16.9)	261 (26.2)	<0.0001
14	192 (19.0)	182 (18.3)	
15	261 (25.9)	148 (14.9)	
≥16	385 (38.2)	404 (40.6)	
Breast feeding			0.488
No	103 (10.3)	90 (9.00)	
Yes	896 (89.7)	916 (91.0)	
Abortion			<0.0001
No	411 (41.8)	261 (26.1)	
Yes	573 (58.2)	738 (73.9)	
Menopause			<0.0001
No	254 (26.0)	348 (34.6)	
Yes	724 (74.0)	659 (65.4)	
Estrogen replacement therapy			<0.001
No	831 (96.6)	895 (92.5)	
Yes	29 (3.4)	73 (7.5)	
Birth control pills			0.156
No	810 (87.3)	818 (85.0)	
Yes	118 (12.7)	144 (15.0)	
History of benign breast diseases			
No	699 (72.0)	640 (64.8)	<0.001
Yes	272 (28.0)	348 (35.2)	
History of breast cancer in first-degree relatives			
No	935 (97.1)	892 (93.7)	<0.001
Yes	28 (2.9)	60 (6.3)	
TNM			
Early stage (0-IIA)	/	579 (64.2)	
Late stage (IIB-IV)	/	323 (35.8)	
Molecular Subtype			
Luminal A	/	575 (56.5)	
Luminal B	/	179 (17.6)	
Her2+	/	180 (17.7)	
Basal-like	/	84 (8.2)	
AGEs (ng/mL)			<0.0001
Q1 (<=1.714)	367 (36.0)	143 (14.0)	
Q2 (1.715–4.404)	267 (26.0)	243 (23.9)	
Q3 (4.405–10.241)	214 (21.0)	295 (29.0)	
Q4 (≥10.242)	173 (17.0)	337 (33.1)	
Continuous	2.6 (1.1, 7.4)	6.5 (3.0, 12.7)	<0.0001
sRAGE (pg/mL)			<0.0001
Q1 (≤557.728)	197 (19.3)	313 (30.7)	
Q2 (557.729–742.104)	231 (22.6)	279 (27.4)	
Q3 (742.105–987.953)	274 (26.8)	234 (23.0)	
Q4 (≥987.954)	319 (31.2)	192 (18.9)	
Continuous	826.1 (606.1, 1050.6)	682.5 (522.3, 914.7)	<0.0001
AGEs/sRAGE (* 1000)			<0.0001
Q1 (≤2.100)	376 (36.8)	134 (13.2)	
Q2 (2.101–5.820)	281 (27.5)	229 (22.5)	
Q3 (5.821–15.129)	211 (20.7)	298 (29.3)	
Q4 (≥15.130)	153 (15.0)	357 (35.0)	
Continuous	3.4 (1.3, 9.3)	9.3 (4.0, 21.1)	<0.0001

AGEs/sRAGE (* 1000) presents a thousand times the ratio of AGEs and sRAGE. Abbreviations: AGEs, advanced glycation end-products; BMI, body mass index; HER2, human epidermal growth factor receptor 2; SD, standard deviation; sRAGE, soluble receptor for advanced glycation end-products; TNM, tumor node metastasis.

**Table 2 cancers-14-06124-t002:** Distribution of AGEs and sRAGE by baseline characteristics.

	AGEs(ng/mL)				*p*	sRAGE(pg/mL)				*p*
	Q1	Q2	Q3	Q4		Q1	Q2	Q3	Q4	
BMI					0.002					<0.001
≤23.9	250 (49.4)	237 (46.8)	217 (43.4)	187 (37.0)		170 (33.5)	210 (41.4)	219 (43.5)	292 (58.3)	
24–27.9	194 (38.3)	194 (38.2)	196 (39.2)	229 (45.2)		239 (47.1)	208 (41.0)	205 (40.8)	161 (32.1)	
≥28	62 (12.3)	75 (14.8)	87 (17.4)	90 (17.8)		98 (19.3)	89 (17.6)	79 (15.7)	48 (9.6)	
Education					0.001					0.022
under primary school	40 (8.0)	45 (9.0)	73 (14.8)	67 (13.8)		73 (15.0)	59 (11.9)	51 (10.3)	42 (8.4)	
Junior/Senior High School	340 (67.7)	361 (72.1)	315 (63.9)	332 (68.3)		319 (65.4)	349 (70.2)	337 (67.9)	343 (68.5)	
Junior college or above	122 (24.3)	95 (18.9)	105 (21.3)	87 (17.9)		96 (19.6)	89 (17.9)	108 (21.8)	116 (23.2)	
Smoking					0.010					0.027
No	460 (93.3)	449 (90.9)	422 (87.0)	433 (90.0)		441 (89.8)	455 (93.0)	444 (91.0)	424 (87.4)	
Yes	33 (6.7)	45 (9.1)	63 (13.0)	48 (10.0)		50 (10.2)	34 (7.0)	44 (9.0)	61 (12.6)	
Drinking					0.559					0.490
No	478 (96.0)	475 (97.1)	467 (96.7)	470 (97.5)		472 (96.7)	477 (97.2)	465 (95.9)	476 (97.5)	
Yes	20 (4.0)	14 (2.9)	16 (3.3)	12 (2.5)		16 (3.3)	14 (2.8)	20 (4.1)	12 (2.5)	
Negative events					0.050					0.091
No	408 (82.9)	382 (78.9)	378 (77.9)	357 (75.8)		373 (77.2)	369 (76.7)	381 (79.0)	402 (82.7)	
Yes	84 (17.1)	102 (21.1)	107 (22.1)	114 (24.2)		110 (22.8)	112 (23.3)	101 (21.0)	84 (17.3)	
Menopause					0.017					0.018
No	130 (21.6)	150 (24.9)	148 (24.6)	174 (28.9)		170 (28.2)	165 (27.4)	130 (21.6)	137 (22.8)	
Yes	371 (26.8)	350 (25.3)	342 (24.7)	320 (23.1)		332 (24.0)	332 (24.0)	364 (26.3)	355 (25.7)	

Abbreviations: AGEs, advanced glycation end-products; BMI, body mass index; sRAGE, soluble receptor for advanced glycation end-products.

**Table 3 cancers-14-06124-t003:** Odds ratios (ORs) and 95% confidence intervals (CIs) for risk of breast cancer associated with quartiles of AGEs-RAGE axis stratified by BMI, menopause, and age.

	BMI ^1^			*p* for Interaction	Menopause ^2^		*p* for Interaction	Age ^3^		*p* for Interaction
	≤23.9	24–27.9	≥28		No	Yes		<60	≥60	
AGEs (ng/mL)				0.898			0.070			0.390
Q1	ref				ref			ref		
Q2	2.827 (1.776–4.502)	2.512 (1.505–4.193)	1.337 (0.572–3.124)		3.723 (1.899–7.299)	2.159 (1.478–3.154)		2.830 (1.992–4.021)	1.339 (0.621–2.887)	
Q3	4.371 (2.694–7.094)	4.158 (2.469–7.001)	2.857 (1.175–6.952)		8.718 (4.325–17.575)	3.002 (2.044–4.408)		4.586 (3.192–6.590)	2.179 (0.985–4.822)	
Q4	5.744 (3.446–9.574)	5.049 (3.018–8.446)	5.738 (2.320–14.195)		8.879 (4.566–17.267)	4.898 (3.267–7.342)		6.155 (4.234–8.947)	3.203 (1.482–6.924)	
sRAGE (pg/mL)				0.533			0.947			0.019
Q1	ref				ref			ref		
Q2	0.562 (0.338–0.936)	0.931 (0.584–1.484)	0.502 (0.212–1.189)		0.705 (0.401–1.240)	0.694 (0.469–1.028)		0.631 (0.445–0.895)	1.337 (0.638–2.803)	
Q3	0.486 (0.295–0.802)	0.732 (0.453–1.184)	0.293 (0.127–0.676)		0.516 (0.281–0.947)	0.548 (0.374–0.804)		0.445 (0.313–0.632)	1.492 (0.699–3.182)	
Q4	0.321 (0.198–0.523)	0.554 (0.332–0.924)	0.307 (0.120–0.786)		0.456 (0.251–0.827)	0.393 (0.265–0.582)		0.406 (0.285–0.578)	0.464 (0.210–1.023)	
AGEs\ sRAGE (* 1000)				0.400			0.524			0.141
Q1	ref				ref			ref		
Q2	2.474 (1.561–3.921)	2.410 (1.417–4.100)	1.031 (0.437–2.432)		2.864 (1.462–5.610)	1.921 (1.311–2.815)		2.578 (1.808–3.676)	1.001 (0.460–2.179)	
Q3	4.679 (2.855–7.667)	3.087 (1.861–5.119)	4.228 (1.728–10.344)		5.783 (2.904–11.517)	3.203 (2.179–4.708)		4.125 (2.879–5.909)	2.665 (1.184–5.997)	
Q4	7.412 (4.442–12.369)	7.276 (4.196–12.619)	6.055 (2.456–14.926)		11.302 (5.775–22.116)	6.396 (4.210–9.718)		8.538 (5.809–12.550)	3.280 (1.484–7.249)	

Model ^1^ adjusted education, menarche age, estrogen replacement therapy, smoking, drinking, negative events, benign breast disease, family history of breast cancer, and menopausal status. Model ^2^ adjusted education, menarche age, estrogen replacement therapy, smoking, drinking, negative events, benign breast disease, family history of breast cancer, and BMI. Model ^3^ adjusted education, menarche age, estrogen replacement therapy, smoking, drinking, negative events, benign breast disease, family history of breast cancer, BMI, and menopausal status. Abbreviations: AGEs, advanced glycation end-products; CI, confidence interval; sRAGE, soluble receptor for advanced glycation end-products.

## Data Availability

The data used for this analysis can be made available upon reasonable request to the corresponding authors.

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
