# Peer review of "Genetically Modified Circulating Levels of Advanced Glycation End-Products and Their Soluble Receptor (AGEs-RAGE Axis) with Risk and Mortality of Breast Cancer"

_cancers, 2022, doi:10.3390/cancers14246124_

Round 1
Reviewer 1 Report
The Authors present a case-control study on the evaluation of the relationship between levels of AGEs-RAGE, their genetic polymorphisms and breast cancer, in 1051 pairs of age-matched breast cancer and controls. They found significant association between higher levels of AGEs and AGEs/sRAGE-ratio and higher risk of breast cancer (BC), with sRAGE levels being negatively associated with breast cancer risk, especially in women <60 years. However, no genetic variants in these genes was associated to an increased risk of BC.
Although the study was properly conducted in terms of plasma analysis, it is not clear if they included patients with a clear hereditary BC condition (i.e. mutated in any of the BC-predisposing genes), because this could be a relevant information since the increased levels of AGEs seems more relevant in younger patients. Morevover, the case-control study on the polymorphisms per se can be flwaed by sdeverasl issues: patient stratification, relatively small sample size. Since data on SNP analysis in large cohort of BC samples are published, it would be interesting to compare the SNP frequencies in those data with the one in the present study.
In conclusion, the data on AGE plasma levels are interesting but need a more detailed analysis to be published in a highly reputed journal.
Author Response
- Although the study was properly conducted in terms of plasma analysis, it is not clear if they included patients with a clear hereditary BC condition (i.e., mutated in any of the BC-predisposing genes), because this could be a relevant information since the increased levels of AGEs seems more relevant in younger patients.
Author response:
We thank the reviewer for catching this important point. However, we regretted that the mutations in BC-predisposing genes such as BRCA have not been detected in this study to exclude hereditary BC patients. According to literature, the mutation rate of BRCA gene in Chinese population is only 5.74% [1], dominated in high-risk groups (early onset, family history of breast cancer and ovarian cancer, bilateral breast cancer) [1]. Just as you mentioned, the increased levels of AGEs seem more relevant to breast cancer risk in younger patients (<60 years old presented in Table 3), thus we performed more cautious stratified analysis by age (Additional Table 1 at the end of this Response letter). We found that early-onset breast cancer (defined as BC patients with a diagnosis age<50 years old) [2] accounted for a large proportion of people younger than 60 years old. AGEs, sRAGE and AGEs/sRAGE ratio in people aged 40-50 years old and 50-59 years old had similar effects on breast cancer risk, and even the Q4 of sRAGE in 50-59 years had a stronger effect than that in 40-50 years, indicating that the difference in sRAGE effect on breast cancer risk between <60 and ≥ 60 years was not entirely caused by genetic factors. In addition, although the effects of AGEs and AGEs/sRAGE-ratio are stronger in people <60 years, they do not interact with age in Table 3 and Additional Table 1.
Furthermore, in this study, we recruited 1, 051 healthy controls who were 1:1 matched to patients by the same age. Although cases and controls may not be completely matched after excluding the undetected and extreme values of AGEs and sRAGE, the age of the two groups was almost the same without statistical difference (Table 1). Also, only newly diagnosed and histologically confirmed unilateral breast cancers were included in our study to eliminate the bilateral breast cancer as much as possible.
Finally, we have adjusted the family history of breast cancer and ovarian cancer in the multivariate logistic regression analysis to control the effect of genetic susceptibility on the association between AGEs/sRAGE and breast cancer.
- Moreover, the case-control study on the polymorphisms per se can be flawed by several issues: patient stratification, relatively small sample size. Since data on SNP analysis in large cohort of BC samples are published, it would be interesting to compare the SNP frequencies in those data with the one in the present study.
In conclusion, the data on AGE plasma levels are interesting but need a more detailed analysis to be published in a highly reputed journal.
Author response:
We greatly appreciate your thoughtful concerns and acknowledge the limitations of the case-control study on the polymorphisms per se. Firstly, our study have actually genotyped the relevant 14 potential functional SNPs located in genes encoding receptors of AGEs in a larger case-control study (10080 controls vs. 8896 breast cancer cases), but only measured plasma AGEs and sRAGE levels in 1051-paired cases and controls due to the relatively expensive costs of the widely approved enzyme linked immunosorbent assay (ELISA) kit (Cell Biolabs, Inc., USA and Quantikine, R&D Systems, MN, USA) we used. Thus, we only reported the findings for these 1051-paired cases and controls with both genotyping and plasma AGEs/sRAGE results. In fact, we have performed the same genotyping analysis in this larger case-control study (10080 controls vs. 8896 breast cancer cases) (Additional Table 2 and 3 shown at the end of this Response letter), and found the consistent results with those shown in this study (Supplementary Table 1 and 5). In addition, we also added the comparison of genotype frequency between SNPs involved in this study and published data in the Results section (Line 362-365, Page 10), showing comparable distributions of the minimum allele frequency of all SNPs involved in this study.
We cannot agree with you more that AGE plasma levels need a more detailed analysis, considering that there are scarce studies on AGEs and breast cancer at present. Further we only explored the relationship between total AGEs and breast cancer, in the future research, we will explore the relationship between different types of AGEs and breast cancer.
References
[1] Gao X, Nan X, Liu Y, Liu R, Zang W, Shan G, et al. Comprehensive profiling of BRCA1 and BRCA2 variants in breast and ovarian cancer in Chinese patients. Hum Mutat. 2020;41(3):696-708.
[2] Ugai T, Sasamoto N, Lee HY, Ando M, Song M, Tamimi RM, et al. Is early-onset cancer an emerging global epidemic Current evidence and future implications. Nat Rev Clin Oncol. 2022;19(10):656-73.
Additional Table 1. Odds ratios (ORs) and 95% confidence intervals (CIs) for risk of breast cancer associated with quartiles of AGEs-RAGE axis stratified by age.
|
|
Age |
P for intera- ction |
||
|
40-49 (N=319) |
50-59 (N=490) |
>=60 (N=209) |
||
|
AGEs (ng/mL) |
|
|
|
0.164 |
|
Q1 |
ref |
|
|
|
|
Q2 |
4.236 (2.251-7.973) |
2.378 (1.535-3.684) |
1.339 (0.621-2.887) |
|
|
Q3 |
6.523 (3.401-12.510) |
3.984 (2.535-6.260) |
2.179 (0.985-4.822) |
|
|
Q4 |
6.699 (3.551-12.638) |
6.418 (3.976-10.360) |
3.203 (1.482-6.924)
|
|
|
sRAGE (pg/mL) |
|
|
|
0.033 |
|
Q1 |
ref |
|
|
|
|
Q2 |
0.590 (0.325-1.073) |
0.676 (0.434-1.055) |
1.337 (0.638-2.803) |
|
|
Q3 |
0.401 (0.216-0.745) |
0.456 (0.293-0.710) |
1.492 (0.699-3.182) |
|
|
Q4 |
0.527 (0.281-0.989) |
0.348 (0.224-0.542)
|
0.464 (0.210-1.023) |
|
|
AGEs\ sRAGE (*1000) |
|
|
|
0.225 |
|
Q1 |
ref |
|
|
|
|
Q2 |
3.282 (1.731-6.223) |
2.292 (1.479-3.553) |
1.001 (0.460-2.179) |
|
|
Q3 |
4.945 (2.647-9.238) |
3.719 (2.364-5.850) |
2.665 (1.184-5.997) |
|
|
Q4 |
8.721 (4.564-16.664) |
9.401 (5.720-15.452) |
3.280 (1.484-7.249) |
|
The minimum age was 44 years.
Covariates: education, menarche age, estrogen replacement therapy, smoking, drinking, negative events, benign breast disease, family history of breast cancer, BMI, menopausal status.
Additional Table 2. Basic information of 14 SNP sites in the total subjects (10080 controls and 8896 breast cancer cases).
|
Gene (Protein) |
SNP locus |
Allele |
MAF |
|
AGER (RAGE) |
rs2071288 |
C/T |
0.01 |
|
rs184003 |
C/A |
0.14 |
|
|
rs2070600 |
C/T |
0.20 |
|
|
rs1800624 |
A/T |
0.16 |
|
|
rs1800625 |
A/G |
0.14 |
|
|
DDOST (AGE-R1) |
rs607254 |
G/A |
0.27 |
|
rs3738140 |
G/A |
0.10 |
|
|
rs4704 |
G/A |
0.39 |
|
|
rs10916846 |
T/C |
0.06 |
|
|
PRKCSH (AGE-R2) |
rs11557488 |
G/A |
0.05 |
|
rs160841 |
A/G |
0.12 |
|
|
LGALS3 (AGE-R3) |
rs1009977 |
T/G |
0.27 |
|
rs4644 |
C/A |
0.19 |
|
|
rs4652 |
A/C |
0.41 |
Additional Table 3. Associations of genetic variants in AGEs-RAGE axis with risk and prognosis of breast cancer in the larger case-control study (10080 controls vs. 8896 breast cancer cases).
|
Gene (Protein) |
SNPs |
Genotype |
Control |
Case |
P |
OR (95%CI) |
HR (95%CI) |
|
N (%) |
N (%) |
||||||
|
AGER (RAGE) |
rs2071288 |
CC |
9827 (97.82) |
8578 (97.26) |
0.012 |
1.00 (reference) |
1.00 (reference) |
|
|
CT |
219 (2.18) |
242 (2.74) |
|
1.200 (0.968-1.486) |
1.241 (0.791-1.948) |
|
|
|
rs184003 |
CC |
7416 (73.88) |
6443 (72.85) |
0.256 |
1.00 (reference) |
1.00 (reference) |
|
|
|
AC |
2426 (24.17) |
2229 (25.20) |
|
1.020 (0.944-1.102) |
1.123 (0.920-1.370) |
|
|
|
AA |
196 (1.95) |
172 (1.94) |
|
0.944 (0.743-1.200)
|
0.625 (0.295-1.323) |
|
|
rs2070600 |
CC |
6384 (63.67) |
5611 (63.67) |
0.949 |
1.00 (reference) |
1.00 (reference) |
|
|
|
CT |
3264 (32.56) |
2862 (32.47) |
|
1.014 (0.944-1.090) |
0.961 (0.793-1.165) |
|
|
|
TT |
378 (3.77) |
340 (3.86) |
|
1.161 (0.975-1.384)
|
1.032 (0.685-1.556) |
|
|
rs1800624 |
AA |
7030 (69.99) |
6167 (69.79) |
0.742 |
1.00 (reference) |
1.00 (reference) |
|
|
|
AT |
2737 (27.25) |
2409 (27.26) |
|
1.009 0.936 1.088 |
1.101 (0.907-1.337) |
|
|
|
TT |
278 (2.77) |
261 (2.95) |
|
1.111 (0.912-1.354) |
1.124 (0.699-1.809) |
|
|
rs1800625 |
AA |
7413 (74.04) |
6519 (73.89) |
0.946 |
1.00 (reference) |
1.00 (reference) |
|
|
|
AG |
2356 (23.53) |
2083 (23.61) |
|
0.982 (0.907-1.062) |
0.928 (0.753-1.145) |
|
|
|
GG |
243 (2.43) |
220 (2.49) |
|
1.044 (0.839-1.298) |
1.230 (0.743-2.038) |
|
DDOST (AGER1) |
rs607254 |
GG |
5394 (53.81) |
4677 (52.89) |
0.439 |
1.00 (reference) |
1.00 (reference) |
|
|
AG |
3935 (39.26) |
3534 (39.96) |
|
1.041 (0.971-1.117) |
0.991 (0.826-1.189) |
|
|
|
|
AA |
695 (6.93) |
632 (7.15) |
|
1.094 (0.958-1.249) |
0.905 (0.644-1.273) |
|
|
rs3738140 |
GG |
8054 (80.32) |
7121 (80.63) |
0.792 |
1.00 (reference) |
1.00 (reference) |
|
|
|
AG |
1862 (18.57) |
1609 (18.22) |
|
0.982 (0.901-1.070) |
0.897 (0.711-1.131) |
|
|
|
AA |
111 (1.11) |
102 (1.15) |
|
1.025 (0.747-1.406) |
0.385 (0.096-1.547) |
|
|
rs4704 |
GG |
3791 (37.80) |
3249 (36.80) |
0.363 |
1.00 (reference) |
1.00 (reference) |
|
|
|
AG |
4741 (47.27) |
4246 (48.09) |
|
1.060 (0.986-1.140) |
0.906 (0.750-1.094) |
|
|
|
AA |
1497 (14.93) |
1334 (15.11) |
|
1.078 (0.974-1.193) |
0.792 (0.601-1.045) |
|
|
rs10916846 |
TT |
8859 (88.56) |
7768 (88.06) |
0.285 |
1.00 (reference) |
1.00 (reference) |
|
|
CT |
1144 (11.44) |
1053 (11.94) |
|
1.062 (0.957-1.179) |
0.794 (0.586-1.075) |
|
|
PRKCSH |
rs11557488 |
GG |
9079 (90.63) |
8042 (90.82) |
0.850 |
1.00 (reference) |
1.00 (reference) |
|
(AGER2) |
|
AG |
911 (9.09) |
791 (8.93) |
|
0.958 (0.852-1.077) |
1.097 (0.804-1.498) |
|
|
|
AA |
28 (0.28) |
22 (0.25) |
|
0.653 (0.341-1.252) |
/ |
|
|
rs160841 |
AA |
7454 (76.71) |
6597 (76.71) |
0.816 |
1.00 (reference) |
1.00 (reference) |
|
|
AG |
2098 (21.59) |
1867 (21.71) |
|
1.015 (0.935-1.102) |
1.146 (0.931-1.409) |
|
|
|
|
GG |
165 (1.70) |
136 (1.58) |
|
0.801 (0.611-1.051) |
1.132 (0.558-2.296) |
|
LGALS3 |
rs1009977 |
TT |
5290 (52.83) |
4756 (54.03) |
0.248 |
1.00 (reference) |
1.00 (reference) |
|
(AGER3) |
|
GT |
3987 (39.81) |
3407 (38.70) |
|
0.960 (0.895-1.029) |
1.013 (0.843-1.218) |
|
|
|
GG |
737 (7.36) |
640 (7.27) |
|
1.002 (0.878-1.143) |
1.148 (0.816-1.613) |
|
|
rs4644 |
CC |
6625 (66.14) |
5859 (66.46) |
0.663 |
1.00 (reference) |
1.00 (reference) |
|
|
|
AC |
3024 (30.19) |
2654 (30.10) |
|
0.995 (0.925-1.071) |
1.019 (0.842-1.233) |
|
|
|
AA |
368 (3.67) |
303 (3.44) |
|
0.983 (0.819-1.181)
|
1.160 (0.737-1.826) |
|
|
rs4652 |
AA |
3555 (35.49) |
3143 (35.68) |
0.428 |
1.00 (reference) |
1.00 (reference) |
|
|
|
AC |
4807 (47.98) |
4272 (48.49) |
|
0.981 (0.912-1.056) |
0.956 (0.788-1.161) |
|
|
|
CC |
1656 (16.53) |
1395 (15.83) |
|
0.941 (0.851-1.041) |
1.173 (0.912-1.510) |
Covariates for OR: age, BMI, education, menarche age, menopause, estrogen replacement therapy, smoking, drinking, negative events, history of benign breast disease, breast cancer history of first-degree relatives.
Covariates for HR: age, BMI, education, income, menopause, smoking, drinking, negative events, TNM stage, molecular subtype, cardiovascular disease, diabetes.
Abbreviations: AGEs, advanced glycation end-products; BMI, body mass index; CI, confidence interval; HR, hazard ratio; OR, odds ratio; RAGE, receptor for advanced glycation end-products; SNPs, single-nucleotide polymorphisms; TNM, tumor node metastasis.
Reviewer 2 Report
This work is not enough contribution and innovation. However, the problem statement and motivation could be stronger or more clearly highlighted.
1. The existing literature should be classified and systematically reviewed, instead of being independently introduced one-by-one.
2. The abstract is too general and not prepared objectively. It should briefly highlight the paper's novelty as what is the main problem, how has it been resolved and where the novelty lies?
3. For better readability, the authors may expand the abbreviations at every first occurrence.
4. The author should provide only relevant information related to this paper and reserve more space for the proposed framework.
5. However, the author should compare the proposed algorithm with other recent works or provide a discussion. Otherwise, it's hard for the reader to identify the novelty and contribution of this work.
6. The descriptions given in this proposed scheme are not sufficient that this manuscript only adopted a variety of existing methods to complete the experiment where there are no strong hypothesis and methodical theoretical arguments. Therefore, the reviewer considers that this paper needs more works.
The algorithm presented has not any novelty.
7. The related works section is very short and no benefits from it. I suggest increasing the number of studies and add a new discussion there to show the advantage.
8. The manuscript is not well organized. The introduction section must introduce the status and motivation of this work and summarize with a paragraph about this paper.
Author Response
Response to Reviewer 2 Comments
Dear reviewers,
We thank you for your efforts in leading the review of our manuscript entitled " Genetically modified circulating levels of advanced glycation end-products and their soluble receptor (AGEs-RAGE axis) with risk and mortality of breast cancer " (cancers-1967989).
We are now pleased to submit our revised manuscript for your consideration. Per editorial board’s suggestions, we have responded to all comments and revised the manuscript, with the changes marked up using the “Track Changes”. Please see below the point-by-point responses to the reviewers’ comments.
All authors have approved the response letter and the revised version of the manuscript. We confirm that there is no image duplication, image manipulation, or visual plagiarism in the manuscript.
We thank you for all your insights, and the opportunity to improve our work. We hope that the revised version of the manuscript is now acceptable for publication in Cancers. We look forward to hearing from you. All correspondence can be sent to my attention.
Reviewer #2: This work is not enough contribution and innovation. However, the problem statement and motivation could be stronger or more clearly highlighted.
Point 1: The existing literature should be classified and systematically reviewed, instead of being independently introduced one-by-one.
Author response:
Thanks for your thoughtful comments. Accordingly, we systematically reviewed and summarized the existing literature and revised previous descriptions of some literature (Line 64-83, Page 2; Line 93-113, Page 2-3; Line 421-431 and Line 437-456, Page 12; Line 459-478 and Line 493-510, Page 13; Line 514-522, Line 534-539 and Line 544-550, Page 14).
Point 2: The abstract is too general and not prepared objectively. It should briefly highlight the paper's novelty as what is the main problem, how has it been resolved and where the novelty lies?
Author response:
Thanks for your comments. We have modified the abstract, highlighted the main problems and how to solve them, and emphasized the novelty (Page 1).
Point 3: For better readability, the authors may expand the abbreviations at every first occurrence.
Author response:
Following your rational suggestions, we have expanded the abbreviations at every first occurrence for better readability.
Point 4: The author should provide only relevant information related to this paper and reserve more space for the proposed framework.
Author response:
We are really appreciated for your thoughtful comments. We have shortened or deleted some unrelated contents in the introduction (Page 2-3) and discussion (Page 12-14), providing space for the relevant contents of this article.
Point 5: However, the author should compare the proposed algorithm with other recent works or provide a discussion. Otherwise, it's hard for the reader to identify the novelty and contribution of this work.
Author response:
Thank the reviewer for catching this important point. In the discussion section (Page 12-14), we added a discussion on the comparison between this study and other studies, highlighting the advantages and novelty of this study.
Point 6: The descriptions given in this proposed scheme are not sufficient that this manuscript only adopted a variety of existing methods to complete the experiment where there are no strong hypothesis and methodical theoretical arguments. Therefore, the reviewer considers that this paper needs more works.
Author response:
We greatly appreciate your thoughtful concerns. At present, there are many measurement technologies for plasma AGEs, among which ELISA is a simple and fast method and the most commonly used technology to detect AGEs in clinical practice [1]. Since there are scarcely any studies on the relationship between AGEs-sRAGE axis and breast cancer, and our research is still in the exploratory stage, we chose this method. In the future, we will further examine the relationship between different types of AGEs and breast cancer. In addition, based on the current research on AGEs or sRAGE, we not only explored the effect of AGEs-RAGE axis on breast cancer, but also evaluated the role of genetic modification by AGEs metabolism-related genes. This is more comprehensive, in-depth and with larger sample size than the existing research on the association between AGEs, sRAGE and breast cancer. However, according to your suggestion, we will improve the research design and experiment more comprehensively in the following studies.
Point 7: The related works section is very short and no benefits from it. I suggest increasing the number of studies and add a new discussion there to show the advantage.
Author response:
Thanks for your comments. According to your suggestions, we have increased the description for the relevant studies and modified the discussion to show our strengths (Page 12-14).
Point 8: The manuscript is not well organized. The introduction section must introduce the status and motivation of this work and summarize with a paragraph about this paper.
Author response:
We are really appreciated for your thoughtful comments. We re-organized this manuscript, and revised the introduction to emphasize the current research status and the purpose of this study (Page 2-3).
Reference
[1] Domenico Corica GP, Monica Curro, Tommaso Aversa, Angelo Tropeano, Riccardo Ientile, Malgorzata Wasniewska Methods to investigate advanced glycation end-product and their application in clinical practice Methods.203:90-102.
Round 2
Reviewer 1 Report
The Authors included additonal data to clarify their findings according to the raised questions. I have no additional issues to discuss and think that this work is worth publishing
Reviewer 2 Report
The paper is finetuned and can be accepted.